# Requirements and Solutions for Robust Beam Alignment in Fiber-Coupled Free-Space Optical Systems

Manuel M. Freitas *, Marco A. Fernandes, Paulo P. Monteiro, Fernando P. Guiomar and Gil M. Fernandes

Instituto de Telecomunicações, Universidade de Aveiro, Campus Universitário de Santiago,
3810-193 Aveiro, Portugal
* Correspondence: manuelfreitas@ua.pt

**Abstract:** The continuous growth of Internet data traffic is pushing the current radio-frequency wireless technologies up to their physical limits. To overcome the upcoming bandwidth bottleneck, Free-Space Optics (FSO) is currently deemed as a key breakthrough toward next-generation ultra-high-capacity wireless links. Despite its numerous advantages, FSO also entails several particular challenges regarding the mitigation of the stochastic impairments induced by turbulence and the strict alignment requirements. One of the main issues of FSO communication systems is the mitigation of pointing errors and angle-of-arrival (AoA) fluctuations, which arise from misalignments induced by atmospheric turbulence and vibrations at the transmitting and receiving stations. A common approach to mitigate the impact of pointing errors is the use of an acquisition, tracking and pointing (ATP) system on one or both ends of the FSO link. In this paper, we present a characterization of the pointing errors and the AoA impact on the power budget of the FSO link to quantify the misalignment impairments. Afterwards, we experimentally demonstrate an FSO link with an ATP mechanism at both ends, managed by a control plane that enables the continuous and accurate alignment of the FSO link. To increase the misalignment tolerance, the ATP mechanism comprises two stages: the first one is based on a spatial diversity method provided by a quadrant detector, while the second stage maximizes the optical received power. Lastly, the impact of the beam misalignment on the achievable information rate of a coherent optical wireless system is theoretically addressed and characterized.

**Keywords:** Free-Space Optics; acquisition; tracking and pointing mechanisms; pointing errors

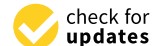



## 1. Introduction

Free-Space Optics (FSO) communication is a line-of-sight (LoS) technology that is attracting significant attention due to its features, such as inherent security, large bandwidth availability, electromagnetic interference immunity and free-licensed spectrum [1]. This kind of optical wireless communication (OWC) has been proposed for several scenarios, which include space (e.g., ground-to-satellite/satellite-to-ground, intersatellite and deep space) and terrestrial communications (e.g., datacenter interconnects, 5G and beyond networks and backup links) [2,3]. FSO communications can be divided into two main categories according to the receiver typologies, which can be based on photodetectors or fiber collimators. The photodetector-based ones tend to have a larger active area facilitating the beam alignment but, consequently, increasing the capacitance of the photodetector constrains their bandwidth. Instead, the fiber-coupled receivers can exploit the compatibility with the currently installed cabling infrastructure and take advantage of the existing mature fiber-optic communication technology, enabling Tbps capacities [1,2]. However, the beam misalignment takes particular relevance in this kind of FSO receiver, in which the light is launched directly into the fiber core.

FSO communication must deal with the challenges associated with atmospheric turbulence and the very high directivity of the optical beam, which may result in beam wandering and pointing errors. This misalignment can also be partially mitigated by using tailored

forward error correction codes [4,5] and advanced modulation techniques, such as the probabilistic constellation shaping (PCS) [6–8]. On the other hand, the traditional way to deal with misalignment comprises the development of highly accurate acquisition, tracking and pointing (ATP) mechanisms. The main task of an ATP mechanism is pointing the transmitter (Tx) in the direction of the receiver (Rx), acquiring the incoming light signal and afterward keeping the FSO link aligned. Thus, the adverse effect of the pointing errors and the angle of arrival (AoA) can be avoided, or partially mitigated, by continuously measuring system-wide performance metrics, such as the received signal power.

An ATP mechanism can be composed of single or several stages using a simple feedback signal based on the received optical power, or it can be assisted by more advanced subsystems that can provide some spatial information about the initial position of the beam. When a given stage ends, the next one will start, until the full beam alignment is concluded. In that way, the stages must play a complementary role. In addition to that, in each stage, a control algorithm steers the beam using a particular feedback response. Hence, the algorithms must also be efficiently designed to partially overcome the limitations imposed by the ATP hardware.

Several ATP mechanisms with different working principles have been proposed, including gimbal-based devices, steering mirrors and wavefront shaping [9]. Each of the aforementioned steering mechanisms has advantages and limitations, and therefore, their standalone or joint use must be properly chosen according to the application scenario. For instance, for long-range transmission systems through the atmosphere, the use of devices that enable the compensation of the fast time-dependent effects induced by beam wandering and scintillation is mandatory, as well as a wavefront compensation subsystem. In contrast, for indoor short-reach transmission systems, the main limitations arise from slow time-dependent deviations of the optical beam.

In this work, we tackle the development of a gimbal-based system able to automatically align a bidirectional FSO link. This control plane will benefit from the constant monitoring and pursuit of a higher power value. The developed multistage algorithm explores, at both ends, alignment processes based on a quadrant detector (QD) sensor feedback for coarse tracking, and an optical-power-based one for fine tracking. The novelty of this paper can be set as follows:

(1) Full characterization and detailed analysis of the pointing errors and the AoA impact on the power budget of an FSO link and on the spectral efficiency of a coherent transmission system.
(2) Detailed description of the developed two-stage bidirectional ATP mechanism: the 1st stage with a power-blind tracking system and the 2nd stage with a gradient descent (GD) algorithm with adaptive power target.

The paper is organized as follows. In Section 2, we review the state of the art and address several research works related to ATP mechanisms in FSO communications. Three methods have received special attention: gimbal-based, mirror-based and wavefront shaping. In Section 3, we present a detailed analysis of the pointing errors and the AoA. In Section 4, we present the ATP system and the developed two-stage alignment algorithm. In Section 5, the spectral efficiency for a coherent transmission system is analyzed. Finally, the main conclusions are presented in Section 6.

## 2. ATP Mechanisms

FSO systems with fiber-coupled optical heads still face limitations in terms of high-precision beam alignment, since the use of passive fiber collimators imposes pointing errors and strongly depends on AoA variations. To mitigate this limitation and increase the reliability and robustness of FSO links, several ATP mechanisms have been developed [6]. These mechanisms can be classified according to their working principles, which have been broadly grouped as gimbal-based, mirror-based and wavefront-shaping (adaptive optics and liquid crystal) solutions; see Figure 1. In addition to these mechanisms, other beam-steering techniques have recetnly been proposed, resorting to optical phased arrays

and decentered variable focus lenses (VFLs) to aid in the tracking and pointing of the optical beam [10]. Moreover, integrating the different aforementioned mechanisms can lead to a more robust ATP system [9]. In this section, we review the existing ATP mechanisms and discuss the main innovative work concerning these methods: the different ATP mechanisms, alignment algorithms and how the algorithms complement the ATP actuator's performance.

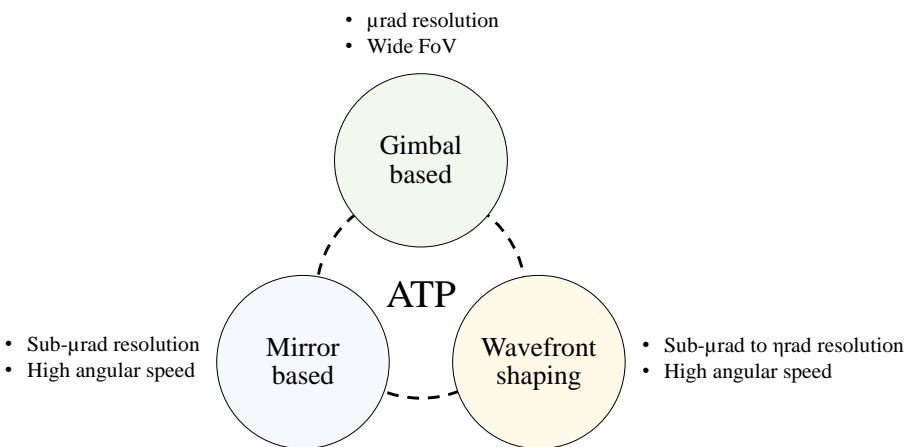

**Figure 1.** Classification of ATP mechanisms in FSO communications according to their working principle and use cases.

## *2.1. Gimbal-Based*

Gimbal-based ATP mechanisms use a mechanical rotary gimbal controlled by motors and have motorized platforms that provide two- or three-axis moving capability to the gimbal. The two-axis one is typically used for applications that require rotating objects in different angular positions at specific angular speeds. The gimbal rotates the mounted object according to the user-desired azimuth and elevation specifications. However, one of the downsides of a two-axis gimbal is the lock that occurs when its rotators reach the limits of their ability to move within the gimbal detection mechanism, i.e., they hit hard stops that prevent further movement. The three-axis gimbal can be a solution since it has one more axis of rotation [11]. These mechanisms are the preferred fit for transceivers that require a wide range of field-of-view (FoV). However, its coarser pointing resolution (i.e., large step size) may be a limitation, especially in cases where fine-grained tracking is sought. If the gimbal mechanism is assisted by another secondary beam-steering system, this limitation can be easily overcome, keeping the advantage of a wide FoV, which is a key limiting factor in its counterpart technologies [9]. It should be noted that recent advances in high-precision electromechanical systems enable to address the needs of FSO systems efficiently. Moreover, the wide FoV makes the gimbal-based systems ideal for multiple input multiple output (MIMO) FSO systems, allowing them to switch between receivers that can be several meters apart [9,11]. These mechanisms are mostly suitable for applications where the weight is not a constraint. For example, small unmanned aerial vehicles (UAVs) require low-mass terminals, and therefore specific light-weight ATP mechanisms. In contrast, large UAVs, trains, cars and satellites can comprise the bulky motors that normally rotate the gimbal [9].

For instance, in [6], an example of a gimbal-based setup is proposed for enabling automatic beam alignment between two FSO transceivers and a fail-safe mechanism for fronthaul links. The authors experimentally demonstrate two alignment methods, artificial intelligence (AI) and a tailored three-stage greedy algorithm (optical-power-based), both relying on a gimbal-based ATP mechanism. This mechanism is composed of two stepper motor actuators mounted on the Tx side, responsible for aligning the respective collimator with a fixed Rx. For scenarios where there is no information about the system, i.e., the beam convergence starts from a blind position, the developed AI method showed an alignment success rate of >96%. On the other hand, for scenarios with partial information about the system, the convergence was faster, although the success rate was lower—92%.

Furthermore, in [12], the authors developed a tracking algorithm that steers the gimbals to maintain full connectivity between two vehicles and consequently enable high-speed communications between both ends of the FSO link. The used 2-axis gimbal was composed of a stepper motor and a position-sensing diode (PSD), in order to allow for tracking and alignment purposes. The development of a gimbal mechanism for space applications based on piezoelectric motors is explored in [13]. The gimbal mechanism has two degrees of freedom, azimuth and elevation directions, and offers low complexity and a high resolution for each step, since it does not need gears nor lubricants and can keep a moderate holding torque, even unpowered. Therefore, this gimbal-based technology is considered very promising for high-resolution pointing applications.

### 2.2. Mirror Based

Mirror-based ATP mechanisms use fast-steering mirrors (FSMs) to perform beam stabilization, pointing and tracking. The optical signal is launched against the mirror, and therefore, by properly adjusting the tip and tilt of the mirror, the reflected optical beam can be oriented in space, enabling the development of a tracking and pointing mechanism. These mechanisms can be developed by using standard mirrors attached to a holder and actuated by a motor or voice coils, the so-called mechanical systems. According to the size of the mirror, they can have considerable moving masses, making them slow, heavy and susceptible to shock and vibration damage. In that way, the performance of the mechanical FSM-based ATP mechanism depends on the size of the mirror: for larger mirrors the systems tend to have lower steering speeds, while for smaller ones they can perform beam realignment in millisecond time scales, with radial accuracies in the range of the µrad [14]. This higher bandwidth can be employed for the constant equalization of the turbulence-induced fading.

An alternative FSM based on piezoelectric and micro-electromechanical systems (MEMS) has been developed with some additional advantages. The piezoelectric FSM provides a large steering range and high precision, but it tends to have a strong nonlinearity response and hysteresis, requiring the use of a complex closed-loop controller. In addition, when compared with the mechanical FSM, it has low power efficiency. The MEMS-based FSM shows similar performance in terms of angular accuracy, it can be divided into electrostatic and electromagnetic actuation kinds. Despite these devices requiring a control loop, they show a quasi-linear response, enabling their use in an open loop with high resolution.

As an application example, in [15], the authors propose different control methods for accurate beam positioning in FSO communication systems. The presented controllers, proportional integral differential (PID) and fuzzy logic, track and analyze the beam position from a QD sensor and posteriorly generate the signal outputs to steer the beam with an FSM. Both systems can successfully align a free-space link, the fuzzy logic controller being the better option compared with the PID controller. On the other hand, in [16], for aiming laser beam machining, the design and tracking control of a high-bandwidth large-aperture FSM actuated by piezoelectric actuators is reported. For the elimination of the piezoelectric hysteresis, an inner digital charge control loop was used, allied with an outer displacement loop to eliminate the resonance. Moreover, feed-forward compensation was applied to reduce the phase lag. This control solution FSM was validated and reached a bandwidth of 10 kHz, with a resolution of <0.3 µrad and precision tracking of maximum radius error <2.2 µrad below 2 kHz for circular trajectory at a radius of 25 µrad. Moreover, in [17], a control system with a PID to improve the working bandwidth of a MEMS piezoelectric FSM is also presented. The piezoelectric actuators are fabricated with scandium-doped aluminum nitride (AlScN) film, which can improve the linearity within small displacement, high-characteristic frequency, low delay, no hysteresis and creep effect. Results show this closed-loop method is able to reduce the overshoot from 71.4% to 8.57%, and the used feedback strategy can reduce the settling time from 398 ms to 2.5 ms.

### 2.3. Wavefront Shaping

2.3.1. Adaptive Optics

Adaptive Optics (AO) is a technology mainly used for atmospheric turbulence compensation based on wavefront sensing and reconstruction. Beyond that, it can also be used for compensating small misalignments of the optical beams, which can be induced by pointing errors or AoA fluctuations. AO enables the mitigation of the wavefront distortions induced by atmospheric turbulence, using deformable mirrors (DMs) and wavefront sensors. For example, a part of the received beam can be directed to a wavefront sensor, reconstructed from the measured data, and afterward, used to calculate control signals for the wavefront corrector actuators [18].

In [19], the AO system was used for orbital angular momentum (OAM) generation and compensation of the atmospheric turbulence degradation, using a Gaussian beam for wavefront sensing and correction at both ends of the bidirectional link. A post- and precompensation of the received/transmitted beam are carried out, reducing the crosstalk between adjacent modes by more than 12 dB, thus proving to be a helpful method for high-capacity FSO bidirectional links. Following the same atmospheric turbulence problem, in [20], an AO mechanism is applied to improve the link performance in coherent FSO communications over maritime atmospheric scenarios. The AO system consisted of three components, the Shack–Hartmann sensor, the control system and the DM. This system was able to optimize the designed closed loop, which helped to successfully correct the wavefront phase distortion and, consequently, enhance the bit-error-rate (BER) performance. More recently, in [21], with the aim of investigating the performance of high-altitude platforms (HAPS)-based FSO communications links, including HAPS-to-ground station (downlink), ground-to-HAPS (uplink) and HAPS-to-HAPS (horizontal link) communications, AO spatial correction filters were analyzed for atmospheric conditions and several parameters, such as beam waist, zenith angle, HAPS altitude, the height of the ground station and receiver aperture diameter, resulting in a considerable improvement in the performance of the three links.

2.3.2. Liquid Crystal Based

Liquid-crystal (LC)-based ATP mechanisms use nonmechanical, fine-beam-steering devices that consist of a one-dimensional array of tens of thousands of long thin electrodes to manipulate the amplitude or phase pattern of a light beam. These devices offer advantages over spatial light modulators (SLMs) and conventional DMs, such as low power consumption, low-cost, light weight and the ability to agilely redirect steering elements. The main advantage of LC-based ATP mechanisms comprises their sub-µrad steering resolution, which is an outstanding feature over other beam-steering devices, such as FSMs. However, this mechanism's major disadvantage is its limited angular motion range [22].

A dual LC-SLM adaptive optics system is used to correct the wavefront distortion of a signal beam under different atmospheric turbulence conditions, and the Strehl ratio (SR) is used as the evaluation index [23]. This type of system was able to achieve phase modulation through the change in the refractive index; moreover, it attracted widespread attention due to its low cost, high resolution, lack of cross-linking between LC molecules, large number of pixel units and ability to independently program and control each pixel [24]. The LC-SLM system was shown to be able to correct the wavefront and increase the SR, as well as turn the corrected spot to be more convergent and enhance the central light intensity. Furthermore, in [25], the necessity to overcome the scattering was reinforced: the refractive distortions and other extreme conditions that turn beam tracking and wavefront sensing into a difficult task. For this, the used SLM technique enabled to realize a huge variety of structured beams and therefore to adapt geometrical parameter flexibility even to difficult measuring situations. This technique demonstrated the ability to enhance wavefront sensing and consecutively simplify beam tracking.

## 3. Misalignment Impairments

The main challenge in the development of a fully autonomous alignment system for the joint optimization of the Tx and Rx results from the unuseful power feedback received on each side of the FSO link if the initial pointing error is too high. In these scenarios, the alignment process might be very challenging due to the absence of a measurable received optical power, making the search for such value very slow and compromising the success rate of the ATP mechanism. However, when a given optical power threshold is found, a proficient algorithm will quickly converge to the optimum system alignment in just a few iterations. This section is devoted to the performance constraints imposed by pointing errors and AoA in an optical-power-based ATP system with seamless fiber–FSO links.

### 3.1. Misalignment Tolerance

To quantify the misalignment tolerance of an FSO system, we designed an experimental setup that allows to control the beam position on the Rx and Tx side in order to evaluate the impact of the pointing errors and the AoA, respectively. The experimental setup consists of a continuous wave (CW) laser that emits 0 dBm and works at a wavelength of 1550 nm, which is connected through a patch-cord of a standard single-mode fiber (SSMF), with a core diameter of 8 µm and an NA of 0.14 to the Tx collimator (Thorlabs F810APC-1550). This collimator ahs two high-precision stepper motors (Thorlabs ZST206) attached, which allow controlling the optical beam position in the y- and x-axis. After traveling through a 3 m FSO channel, the signal is collimated into an SSMF by an analogous collimator, and finally, the optical power is measured. The control and measuring loop works with the optical power values registered by the HP-8157A power meter (PM) connected to a laptop running a MATLAB script; see Figure 2.

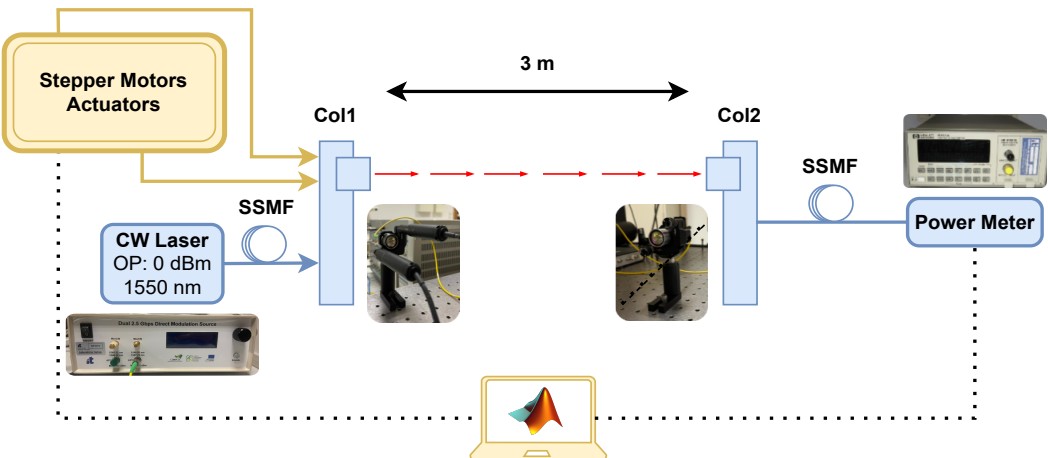

**Figure 2.** Schematic of the experimental setup used for pointing errors characterization. Color codes: blue—optical components; yellow—control and measuring plane.

### 3.1.1. Pointing Errors

By synchronizing the beam movement with a real-time power meter, a 2D sweep of the beam at the col2 plane is performed; see Figure 2. The contour map, for a 12 mm horizontal and vertical displacement, can be observed in Figure 3a. Figure 3b presents a cut of the power contour map, highlighting the width on both axes. Considering a power threshold of −3 dB, the tolerated beam misalignment is about 0.67 mm, while for a power threshold of −20 dB, it increases to 1 mm.

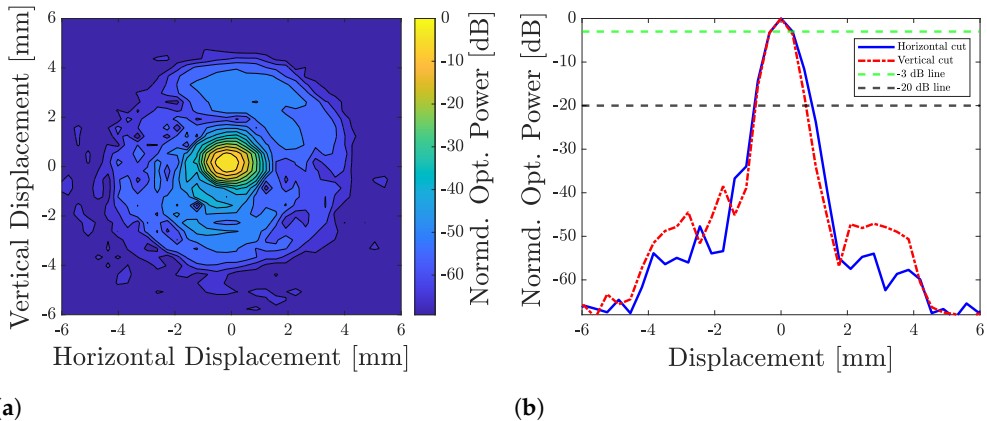

(**a**)                 (**b**)

**Figure 3.** (**a**) SSMF contour map of power received at col2 for the different beam focus. (**b**) SSMF respective power penalty of beam misalignment.

### 3.1.2. Angle of Arrival

The sweep was repeated by moving the col1, however, with the optical beam coming from the fixed col2 (see Figure 4), in order to understand if the impact of the AoA would be similar to the pointing errors; see Figure 5a. Figure 5b shows the same Gaussian behavior previously observed. Note that to represent the AoA in angular units, a simple trigonometric relation can be used between the results presented in Figure 5 and the FSO link length ($\theta = tg^{-1}(\frac{Displacement}{Link_{length}})$); see Figure 4. In addition to that, for the same considered power threshold, the tolerated beam misalignment tends to have similar values, with 0.61 mm and 1 mm for $-3$ and $-20$ dB, respectively.

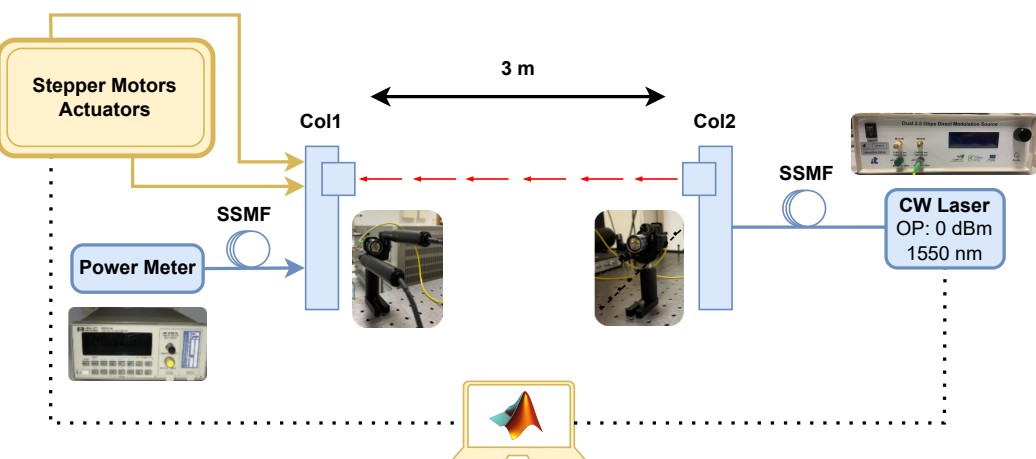

**Figure 4.** Schematic of the experimental setup used for AoA characterization. Color codes: blue—optical components; yellow—control and measuring plane.

In terms of the fiber coupling requirements, the impact of the pointing errors was characterized when the transmitted beam was varied, while the optical power was measured with a fixed receiver; see Figure 3. Conversely, the impact of varying the receiver position while the transmitter is fixed and emitting the optical beam was also measured; see Figure 5. By comparing the optical-power-based contour maps' area for each case, it was possible to observe similar behavior for either Tx- or Rx-induced misalignment. Therefore, we can assume the operating zone for the developed ATP mechanism will be similar, regardless of the side from which the optical beam departs. Since a misalignment as small as 1 mm can induce a power drop of 20 dB, power-based ATP systems must use complementary subsystems to improve their flexibility, operation speed, and reliability. In that way, cameras or QD can be used to assist the ATP system in improving its FoV. These

subsystems can directly detect the optical beam or several beacons placed at the other end of the FSO link.

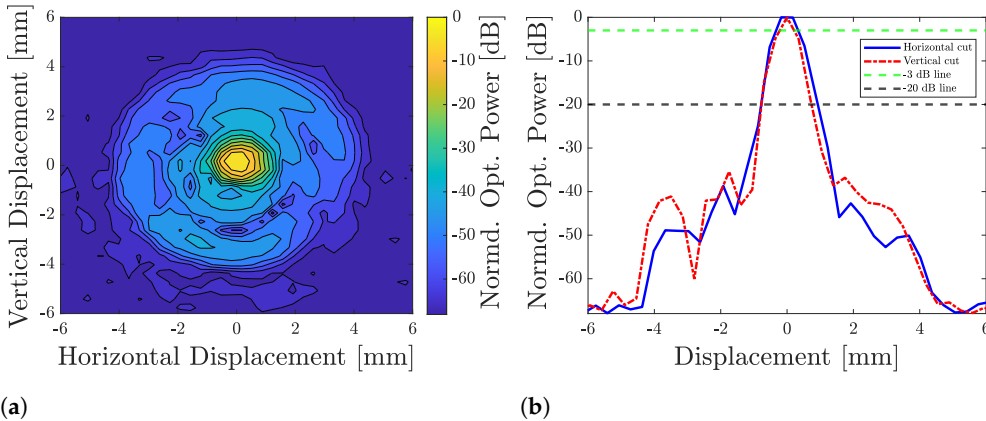

(**a**)                                                                                        (**b**)

**Figure 5.** (**a**) SSMF contour map of power received at col1 for the different beam focus. (**b**) SSMF respective power penalty of beam misalignment.

## 4. Multistage ATP Mechanism

This section describes the architecture of the two-stage algorithm proposed for beam alignment in FSO links. The dual-stage alignment process is based on the QD sensor feedback for coarse tracking and optical power feedback for fine tracking. When the QD-based alignment ends on one side, the control plane starts the coarse alignment on the other side. When the coarse alignment is concluded on both sides, the power-based alignment starts on one side at each time. The schematic of the bidirectional alignment setup is shown in Figure 6.

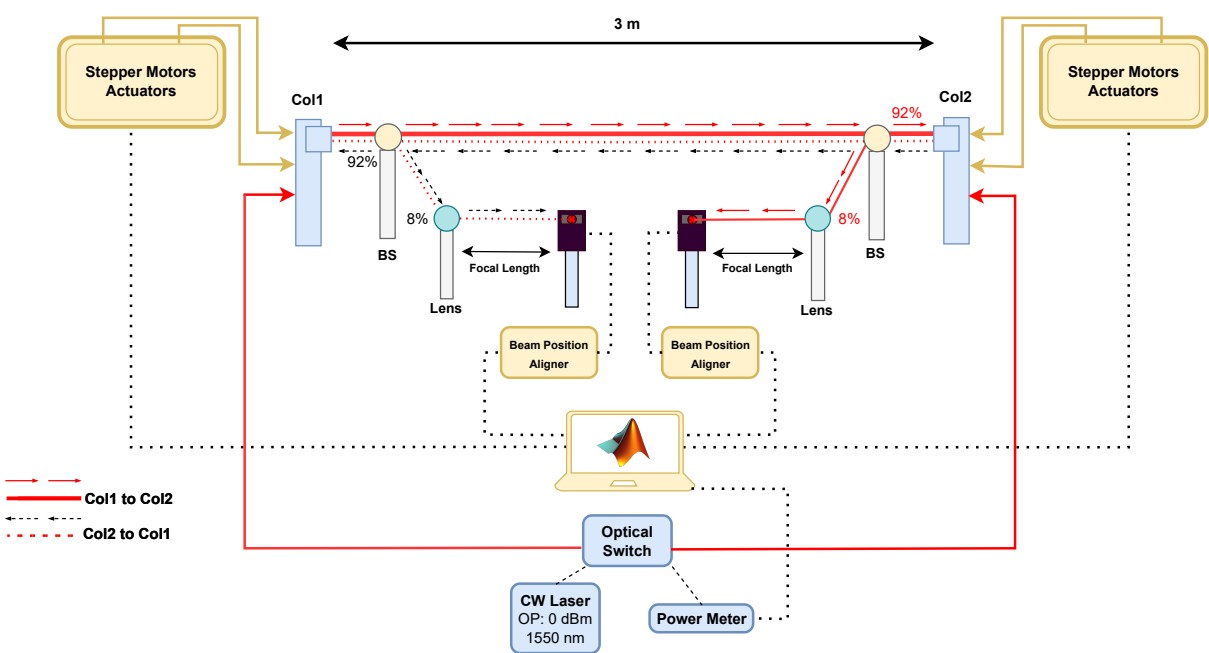

**Figure 6.** Schematic of the bidirectional ATP mechanism developed. Color codes: blue—optical components; yellow—ATP system.

### 4.1. Experimental Setup

For the experimental validation of the proposed ATP mechanism, we used a CW laser emitting 0 dBm at a wavelength of 1550 nm, followed by an optical switch to change the beam direction, i.e., defining whether it is col1 or col2 (Thorlabs F810APC-1550), that is, receiving light from the laser and forwarding it to the PM. For the collimators' tip and tilt

control, two stepper motor actuators (Thorlabs ZST206) are mounted on each collimator. The control and measuring loop work through the received optical power data, registered with an optical PM, and the beam position is tracked by two QD sensors (Thorlabs PDQ30C). Each QD is attached to a beam position aligner (Thorlabs KPA101) that outputs three analog voltages used for beam tracking and pointing: the $X_{\text{diff}}$ and $Y_{\text{diff}}$ signals are proportional to the difference in light detected by the left-minus-right and top-minus-bottom pairs of photodiode elements in the detector array and the signal SUM proportional to the light falling on the sensor. Finally, two beam splitters (BSs) (Thorlabs BP108) are used to reflect 8% of the beam light into the QD sensors, and two Ø1 CaF2 Plano-Convex Lens (Thorlabs LA5817) are used to focus the optical beam incident into the QD. The beam alignment is controlled by a MATLAB script that commands the four stepper motorized actuators to change the tip and tilt of the Tx and Rx collimators via a two-axis collimator mount; see Figure 7. Note that in a real application, the power can be provided directly by the receiver at the output of the collimator with a coupler (with the appropriate ratio), followed by a PM. Moreover, in these scenarios, several radio-frequency (RF) standard technologies, such as LoRa, can be used as a control communication channel to exchange the QD sensor data during the bidirectional alignment.

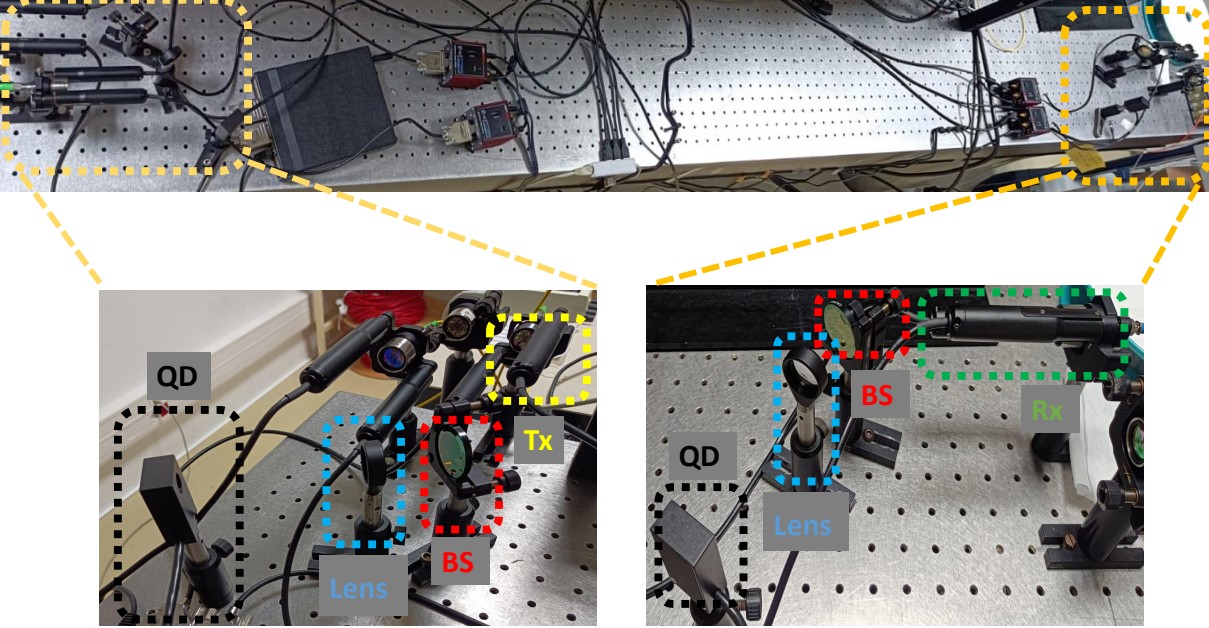

**Figure 7.** Picture of the experimental setup used for bidirectional alignment.

*4.2. Algorithm Architecture*

First, let us note that the developed dual-stage ATP mechanism requires that an initial coarse alignment is performed between the Tx and Rx optical antennas, so that the alignment error is within a few cm from the ideal pointing spot (measured at the receiver plane). This initial coarse alignment can be achieved through different strategies, including computer vision, GPS synchronization, or auxiliary beacon detection (e.g., colocated LEDs or laser diodes) [26]. However, the development of this coarse alignment stage is out of the scope of this paper. Therefore, we assume that such a technology is available and can be seamlessly integrated with our alignment system, providing a starting point for the alignment error that is within the tolerance of the subsequent fine-tuning algorithm that is detailed henceforward. The algorithm architecture exhibited in Figure 8 is described as follows.

1st Stage: This stage is referred to as the QD-based stage. It starts by measuring the signal sum of the voltages produced by each photodiode composing the QD sensor, $V_{\text{QD,SUM}}$. This signal is proportional to the incident light within the sensor, and thus

provides a way to detect if some portion of the received optical beam is being collected within the QD sensor area. By considering a minimum threshold voltage, $V_{th} = 0.2$ V, which leads to the optimum QD response, the algorithm will check if the measured value is sufficient to reliably proceed with the QD-based alignment ($V_{QD,SUM} \geq V_{th}$). Then, if this condition is verified, the stepper motors will move the optical beam to the predefined position given by the coordinates, $V_{QD,x}$ and $V_{QD,y}$, returned by the QD. This position is initially set when calibrating the entire system. During this procedure, only if the measured optical power is greater than or equal to a predefined threshold ($P_{meas} \geq P_{th} = -20$ dB), the 1st stage stops and advances to the next stage. This procedure starts in the Tx-collimator (col1) and then switches to the Rx-collimator (col2). Note that if this power condition is not verified, it means the system needs recalibration.

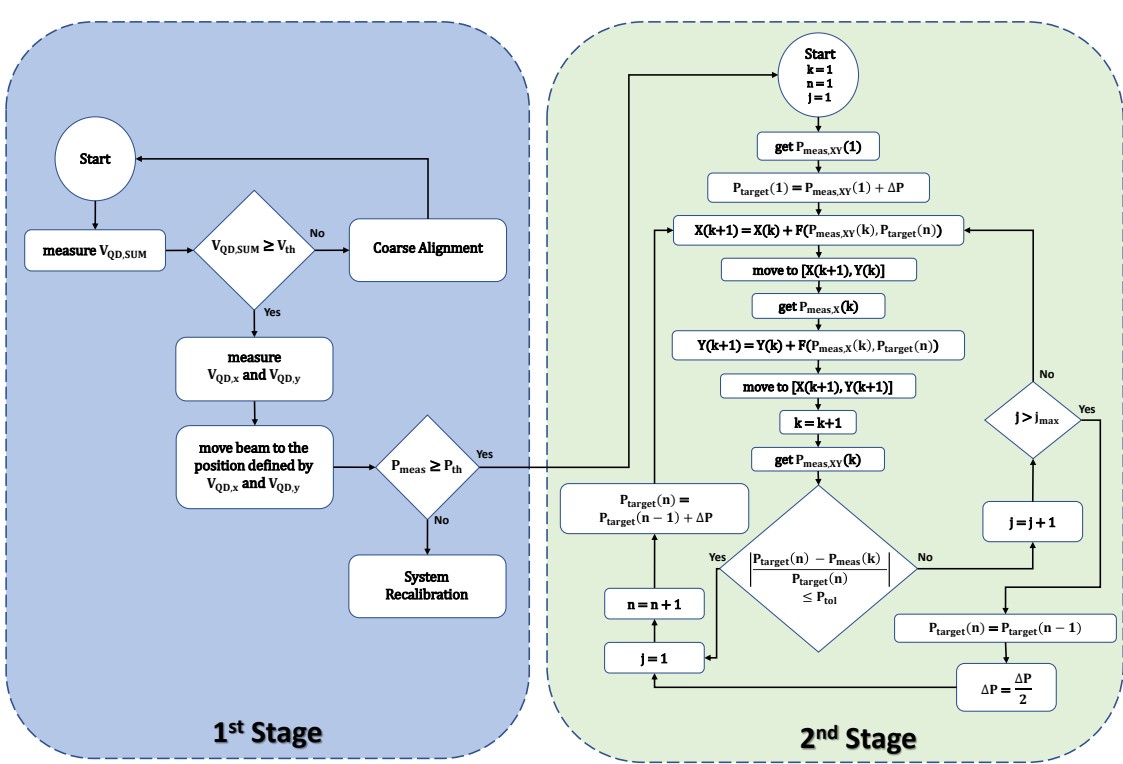

**Figure 8.** Proposed two-stage bidirectional algorithm architecture.

2nd Stage: This stage is referred to as the optical-power-based stage. It starts by measuring the initial optical power that is collimated into the Rx fiber, $P_{meas,XY}(1)$. Afterwards, a power target is defined by incrementing $\Delta P = 0.5$ dB to the previously measured power ($P_{target}(1) = P_{meas,XY}(1) + \Delta P$), and the optimization algorithm is employed to reach that threshold by setting different absolute positions ($X(k)$ and $Y(k)$). After $j$ iterations, the target is expected to be reached, i.e., the error should vanish. Otherwise, the algorithm has entered a saturation zone, $j > j_{max} = 10$, where the set target has exceeded the maximum value that can be reached. In this case, the next $P_{target}$ will be equal to the previous one that allowed the algorithm convergence ($P_{target}(n) = P_{target}(n-1)$). Then, the $\Delta P$ is redefined until the algorithm again enters a saturation zone. At that time, the position after compensation is defined as the new predefined QD position that will be used when further misalignment or power fluctuations affect the FSO link.

Note that for the sake of simplicity, Figure 8 only represents the alignment in one direction. In addition to that, the 1st stage can be performed in parallel, since the AoA impact is not a problem when aligning through the QD sensor. Regarding the 2nd stage, performing it in parallel can be more challenging, since each dithering imposed by the motors will change the pointing errors in one direction and the AoA in the opposite direction.

Adaptive Beam Alignment Using an Optimizer Gradient Descent

The GD algorithm is often used as a black-box optimizer. It is an optimization algorithm used to minimize some functions by iteratively moving in the direction of the steepest descent, as defined by the negative gradient [27]. The gradient at the current position is used to iteratively calculate the next point. Furthermore, it is scaled by a learning rate, and the obtained value is subtracted from the current position (makes a step) in order to minimize the function. In this case, the algorithm was adapted to search for the highest value of optical power. Accordingly to the block diagram of Figure 8, the alignment procedure starts by measuring the average optical power at a starting $[X(k), Y(k)]$ position, $P_{\mathrm{meas},XY}(k)$, based on which the new $X(k+1)$ position is found,

$$X(k+1) = X(k) + F(P_{\mathrm{meas},XY}(k), P_{\mathrm{target}}(n)), \tag{1}$$

where $F$, $P_{\mathrm{meas}}$ and $P_{\mathrm{target}}$ denote the cost function, the currently measured optical power value and the optical power target, respectively. The $P_{\mathrm{meas}}$ value should be an average of multiple measured values, thus reducing the impact of time-varying power fluctuation on the alignment procedure.

Then, a new power measurement, $P_{\mathrm{meas},X}(k)$, is taken after moving the stepper motors only over the $X$-axis to position $[X(k+1), Y(k)]$. Using this updated power measurement, the $Y$ coordinate is updated using the same GD algorithm,

$$Y(k+1) = Y(k) + F(P_{\mathrm{meas},X}(k), P_{\mathrm{target}}(n)). \tag{2}$$

The cost function employed in expressions (1) and (2) is given by

$$F(P_{\mathrm{meas}}(k), P_{\mathrm{target}}(n)) = \mu \, \mathrm{sign}(P_{\mathrm{meas}}(k) - P_{\mathrm{meas}}(k-1)) \left| \frac{P_{\mathrm{target}}(n) - P_{\mathrm{meas}}(k)}{P_{\mathrm{target}}(n)} \right|, \tag{3}$$

where:

- $\mu$ denotes the learning rate. The learning rate is a variable that requires careful choice, since it is directly related to the step size, which has a strong influence on the algorithm performance. If $\mu$ is too small, the algorithm may take a long time to converge or can reach the maximum number of iterations defined by the user before reaching the optimum point. On the other hand, if $\mu$ is too high, the GD can cover more ground, but it can lead to overshooting the lowest point or diverging completely. For reference, we used $\mu = 30$.
- $\mathrm{sign}(P_{\mathrm{meas}}(k) - P_{\mathrm{meas}}(k-1))$ defines the direction of movement over the $X/Y$ axis, where

$$\mathrm{sign}(x) = \begin{cases} 1 & , \ x \geq 0 \\ -1 & , \ x < 0. \end{cases} \tag{4}$$

- $P_{\mathrm{meas}}(k)$ corresponds either to the measured optical power before the application of expression (1), $P_{\mathrm{meas},XY}(k)$ or before the application of expression (2), $P_{\mathrm{meas},X}(k)$.
- the $P_{\mathrm{target}}$ value is calculated using the following process: the function current value will be compared with a previously defined target (starting point) that corresponds to a small increment, $\Delta P$, over the currently measured power value, $P_{\mathrm{meas}}$. This target is adaptive, i.e., once the error is less than or equal to a certain power tolerance ($P_{\mathrm{tol}} = 0.1\,\mathrm{dB}$) during a given number of iterations, its value should increase, and the same process is repeated until the algorithm is not able to converge. At that moment, the previous target that allowed convergence would be set as the current target.

Notice that the working region under analysis is the well-behaved Gaussian zone of the optical power, above $P_{\mathrm{th}}$ (see Figure 9), as it is guaranteed by the 1st alignment stage. This greatly facilitates the reliable convergence of the modified GD algorithm described above.

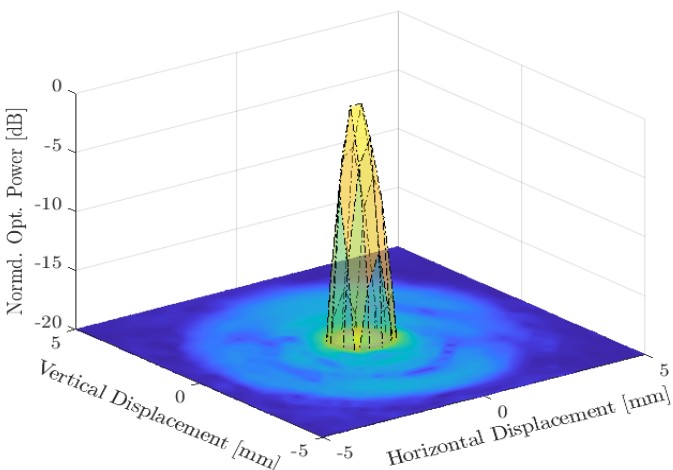

**Figure 9.** Working region for the 2nd-stage optical-power-based alignment, $P_{\text{th}} = -20\,\text{dB}$.

*4.3. Experimental Results*

Figure 10 illustrates the optical power variation while the bidirectional algorithm aligns the FSO link. Starting by moving the Tx-side (col1), the beam trajectory converges to the predefined position at the Rx; see Figure 11a. However, due to the AoA impact, the measured optical power remained at the lower sensitivity level. After reaching that position, the link switches its direction, and the Rx-collimator (col2) alignment also starts converging, thus compensating the AoA and leading to an increase in optical power until reaching a defined threshold; see Figure 11b. At that time, the QD-based stage ends, and a fine-tuning stage based on optical power feedback starts.

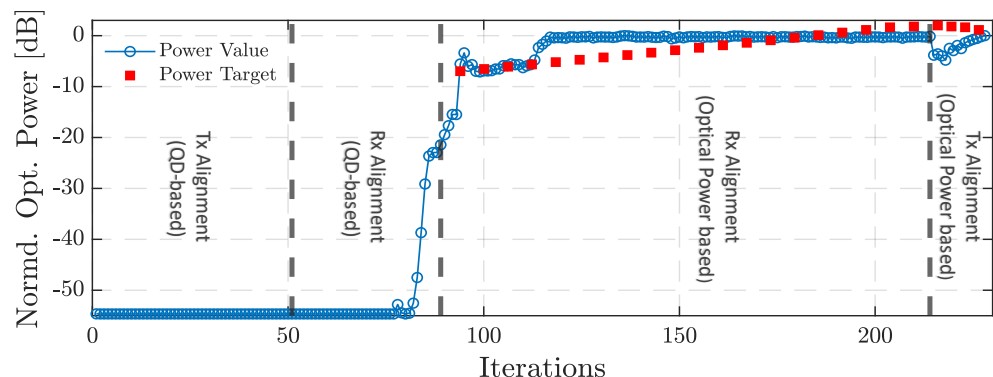

**Figure 10.** Optical power variation during the bidirectional alignment.

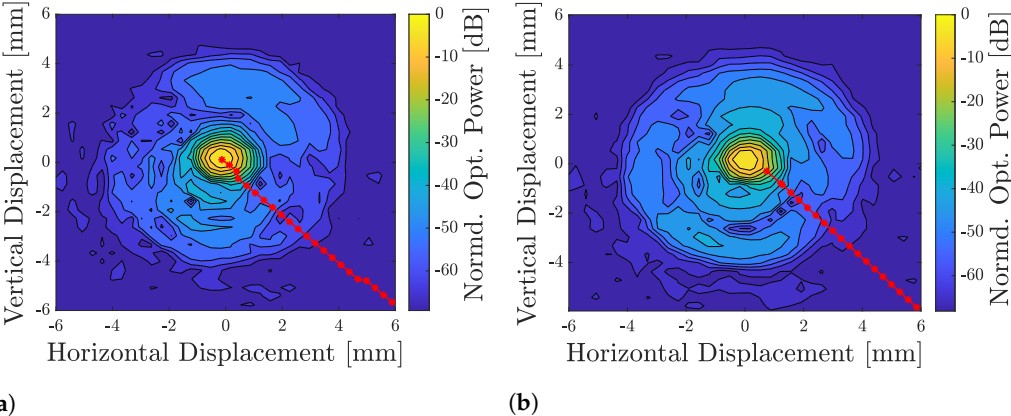

**Figure 11.** QD-based alignment trajectory. (**a**) Tx collimator alignment. (**b**) Rx collimator alignment.

Regarding the 2nd stage, the developed optimization algorithm will play the main role. To be precise, the objective is to compensate for misalignments that can compromise the QD alignment, i.e., the error given by the difference between the adaptive target and the currently measured power. Initially, there is an unstable shooting trying to compensate for the optical error, since the step size is considerably large (see Figure 12a,b) up to the first 20 iterations. From that point on, the error remained nearly zero, since the step size induced to the motors was not enough to trigger a considerable movement. After a certain iteration number with the error remaining in the same optical range, the link direction is switched again. The procedure will be the same; thus, as the power target increases, the algorithm compensation capacity can reach its limit. Analyzing Figure 10, there is a negative power fluctuation at iteration 214 that is justified by the power target being too high. In order to realign the link, the new power target defined is lower, and it is reachable by inducing larger step sizes to the stepper motors; see Figure 12a. This will correspond to higher values of step size, since the motors will need to compensate for the generated divergence and reach the optimal optical power target again (see Figure 12a) after iteration 120. Considering the same iterations, as expected, there is a variation in the optical power error corresponding to each axis that returns to the optimal values as the motors move to the position of interest; see Figure 12b.

The maximum pointing error that this ATP system will be able to compensate will depend on the lens size (Thorlabs LA5817) placed in front of the QD (Thorlabs PDQ30C), i.e., there will be a trade-off between the footprint of the alignment system and the lens size (25.4 mm diameter).

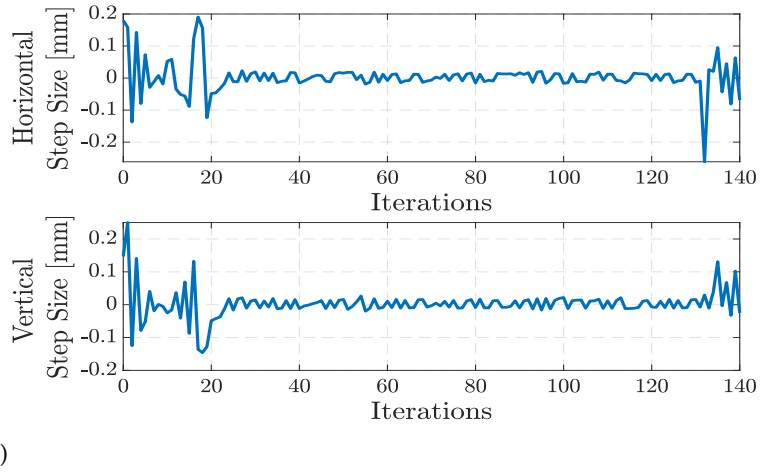

(**a**)

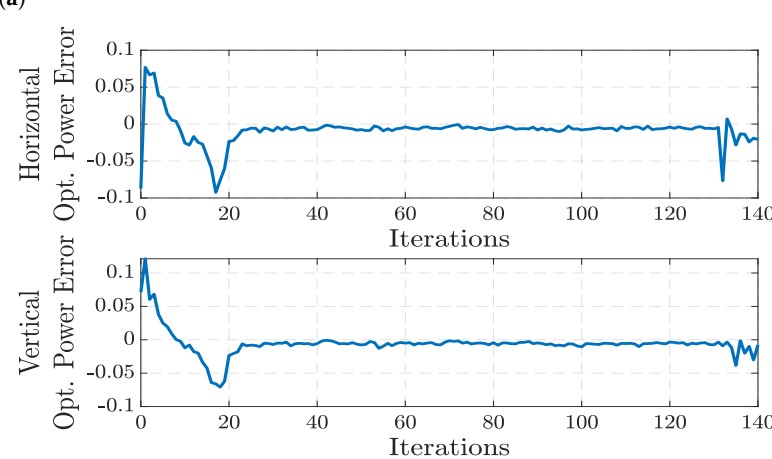

(**b**)

**Figure 12.** (**a**) Registered motors' step size during the 2nd-stage displacement. (**b**) Measured algorithm error during the 2nd-stage displacement.

## 5. Impact of Beam Misalignment on Achievable Information Rate

In order to determine the theoretical highest spectral efficiency for the considered channel, let us consider the Shannon–Hartley theorem [28]:

$$\eta = \frac{C}{B} = \log_2(1 + \text{SNR}) \tag{5}$$

where B is the channel bandwidth, SNR is the signal-to-noise ratio and C is the capacity of the channel in bits per second.

For such analysis, we considered an average SNR of 20 dB, which corresponds to the point with the highest optical power in Figure 3. Considering a linear system, we can assume the drop in optical power corresponds to an identical drop in SNR, since these metrics will be directly proportional, i.e, the Rx noise power will be approximately constant, while the received signal power varies with the beam misalignment. Figure 13a depicts how the spectral efficiency behaves for a displacement of 1.5 mm from the center, i.e., the optimum point. The maximum registered spectral efficiency per polarization is approximately 6.7 bit/s/Hz, which corresponds to 13.4 bit/s/Hz if dual-polarization transmission is employed. Furthermore, it can be seen that after a 0.5 mm displacement, the spectral efficiency reaches values near 4 bit/s/Hz/pol; however, after a 1 mm displacement, the FSO system becomes practically unusable, as the supported spectral efficiency goes to zero; see Figure 13b. This clearly exposes the critical role that is played by optical beam alignment in fiber-coupled FSO systems, operating as an enabling technology to unleash its full potential capacity.

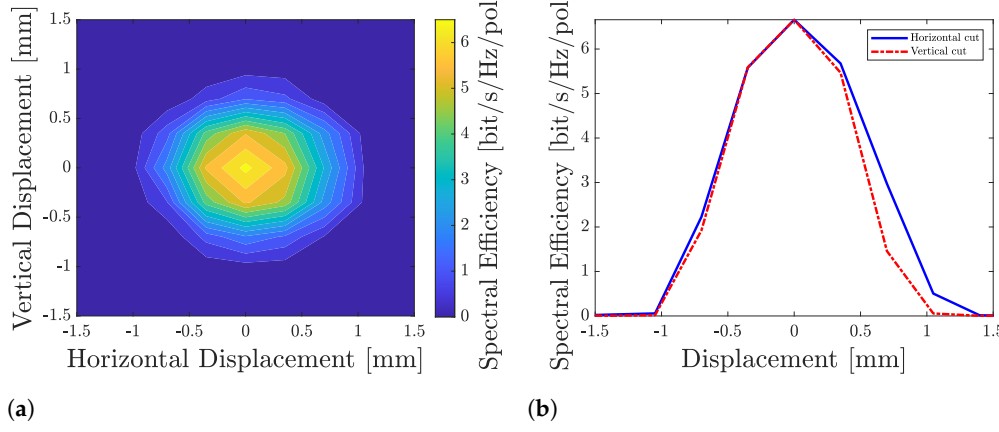

(**a**)                                                 (**b**)

**Figure 13.** (**a**) SSMF contour map of spectral efficiency at the Rx for the different beam focus. (**b**) SSMF respective spectral efficiency of beam misalignment.

## 6. Conclusions

We characterized the impact of the pointing errors in SSMF-coupled systems, observing similar behavior for either Tx- and Rx-induced misalignment, which will enable the ATP mechanism to operate in the same power regions in both directions of the link. Hereupon, a detailed explanation of the developed control plane, which is based on a dual-stage algorithm was given, emphasizing the dependency on aligning with the QD sensor (1st stage) before advancing to an optical-power-based alignment (2nd stage). To guarantee that the initial pointing error is below the QD maximum tolerance, it is necessary to consider a previous alignment stage capable of bringing the beam to the FoV of the sensor. Once the QD-based 1st stage is concluded (i.e., the measured received optical power is above a given predefined threshold), the GD-based optimization algorithm that is the core of the 2nd alignment stage is introduced. This benefits from its computational efficiency and produced stable error gradient and convergence if working in well-behaved zones, i.e., functions without local minima. This mechanism was successfully validated in an indoor scenario. Therefore, it is envisioned that the main target scenario for the proposed ATP system is within the framework of intradatacenter networks, especially due to the required beam



steering between datacenter racks, which may lead to pointing errors [29]. The typical link length can vary from several dozen up to a few hundred meters [30]. According to [31], in a controlled environment similar to the one experienced in a datacenter server room, the turbulence levels are not enough to induce transmission instability in the FSO link. Nevertheless, the inherent flexibility provided by the GD method, associated with the use of average power values resulting from several measurements, allows its use in more challenging environments, where the impact of turbulence is non-negligible. Regarding outdoor turbulent scenarios, the algorithm metrics may require fine-tuning adjustments, namely enabling them to adequately treat the impact of turbulence-induced fading without deteriorating the alignment stability. The optical power measurements during an FSO system bidirectional alignment prove that for any misalignment, if part of the beam is within the QD area, the alignment is possible. Thus, the algorithm ceases to be solely dependent on optical power, becoming more resilient. The subsequent optical-power-based stage is required for fine alignment, most importantly when the initial system calibration conditions are undesirably changed. It will converge to the optimum power value and redefine a new predefined position in the QD sensor. Overall, this work successfully demonstrated a control plane that enables a constant optical beam monitoring and alignment of an FSO link. The system uses a GD algorithm with an adaptive power target, especially beneficial for increasing the power of a system with an unknown link budget. The bidirectional alignment demonstrated high resilience to pointing errors and to the AoA impact. Finally, to highlight the fundamental role of the bidirectional ATP systems presented, the paper closed with a theoretical analysis of the beam misalignment impact on the achievable information rate, reaching an overall spectral efficiency of 6.7 bit/s/Hz per polarization, considering an average SNR of 20 dB at the optimum point.

**Author Contributions:** Conceptualization, M.M.F., M.A.F., F.P.G. and G.M.F.; methodology, M.M.F., M.A.F., F.P.G. and G.M.F.; software, M.M.F.; validation, M.M.F., M.A.F., P.P.M., F.P.G. and G.M.F.; formal analysis, M.M.F.; investigation, M.M.F.; resources, P.P.M. and F.P.G.; data curation, M.M.F.; writing—original draft preparation, M.M.F. and G.M.F.; writing—review and editing, M.M.F., M.A.F., P.P.M., F.P.G. and G.M.F.; visualization, M.M.F., M.A.F., F.P.G. and G.M.F.; supervision, M.A.F., P.P.M., F.P.G. and G.M.F.; project administration, P.P.M. and F.P.G.; funding acquisition, F.P.G. and P.P.M. All authors have read and agreed to the published version of the manuscript.

**Funding:** This work was partially supported by FEDER, through the CENTRO 2020 programme, project ORCIP (CENTRO-01-0145- FEDER-022141), MSCA RISE programme through project DIOR (grant agreement no. 101008280), and by FCT/MCTES through project OptWire (PTDC/EEI-TEL/2697/2021, UIDB/50008/2020-UIDP/50008/2020). Marco A. Fernandes acknowledge PhD fellowship from FCT with code 2020.07521.BD. Fernando P. Guiomar acknowledges a fellowship from "la Caixa" Foundation (ID 100010434), code LCF/BQ/PR20/11770015. Gil M. Fernandes acknowledges a contract program from FCT with code 2022.07168.CEECIND.

**Institutional Review Board Statement:** Not applicable.

**Informed Consent Statement:** Not applicable.

**Data Availability Statement:** Not applicable.

**Conflicts of Interest:** The authors declare no conflict of interest.

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
