# Peer review of "Requirements and Solutions for Robust Beam Alignment in Fiber-Coupled Free-Space Optical Systems"

_photonics, doi:10.3390/photonics10040394_

Round 1

Reviewer 1 Report

The manuscript presents a study on beam alignment in free-space optical communication systems, which remains an important topic despite its broad study. However, there are a couple of comments on the manuscript that need to be addressed.

Firstly, the technical contribution of this study is not presented clearly. Although the authors mention that they use a quadrant detector sensor feedback for coarse tracking and optical power-based methods for fine tracking, which is a common alignment procedure, they need to clarify the novelty of their work. What sets their study apart from previous research, and what are the specific technical contributions that they make?

Secondly, the manuscript contains extensive discussions of various alignment methods, including gimbal-based, mirror-based, and wavefront shaping. It is unclear whether this is intended to be a review paper or a research article. If it is a research article, the authors should focus more on their specific methods and results, and shorten the discussions of other alignment methods. On the other hand, if they intend to deliver a review paper, the introduction should clearly indicate this intention. By doing so, the scope of the paper will become clearer to the reader.

Reviewer 3 Report

Read the attached file.

Round 2

Reviewer 1 Report

This manuscript has been revised and all of the comments provided by the reviewer have been addressed. No further comments have been provided by this reviewer.

Reviewer 3 Report

The additional explanations added to the article are satisfactory and I can recommend the publication of the article in its present form.